# Chronic Complete Distal Aortic Occlusion and Pulmonary Embolism—Atypical Antiphospholipid Syndrome?

**DOI:** 10.3390/diagnostics13071346

**Published:** 2023-04-04

**Authors:** Simona Caraiola, Laura Voicu, Dragoș Cașu, Elena Armășoiu, Claudia Oana Cobilinschi, Emilian Mihai, Răzvan Adrian Ionescu

**Affiliations:** 1Third Internal Medicine Department, Colentina Clinical Hospital, 020125 Bucharest, Romania; 2Fifth Department-Internal Medicine (Cardiology, Gastroenterology, Hepatology, Rheumatology, Geriatrics), Family Medicine, Occupational Medicine, Faculty of Medicine, “Carol Davila” University of Medicine and Pharmacy, 050474 Bucharest, Romania; 3Rheumatology Department, “Sf. Maria” Clinical Hospital, 011172 Bucharest, Romania; 4Cardiology Department, Colentina Clinical Hospital, 020125 Bucharest, Romania

**Keywords:** aortic occlusion, pulmonary embolism, antiphospholipid syndrome, atherosclerosis, thrombosis

## Abstract

Complete aortic occlusion is a rare pathology with various possible etiologies. According to current data, it is most frequently caused by atherosclerosis. However, thrombosis or vasculitis could also be involved. We present the case of a 42-year-old female with chronic complete distal aortic occlusion, associated pulmonary embolism and positive antiphospholipid antibodies. The patient had an obstetric history suggestive of antiphospholipid syndrome (APS). She presented with typical intermittent claudication symptoms persisting for approximately five years at the time of admission. Arteriography revealed complete infrarenal aortic occlusion and the presence of collateral arteries. Aortoiliac bypass surgery was performed. This case emphasizes an unusual, yet possible, etiology of chronic aortic occlusion—most probably, combining atherosclerosis and chronic thrombosis—in a relatively young patient, in which the diagnosis was significantly delayed due to the peculiar association of traits.

**Figure 1 diagnostics-13-01346-f001:**
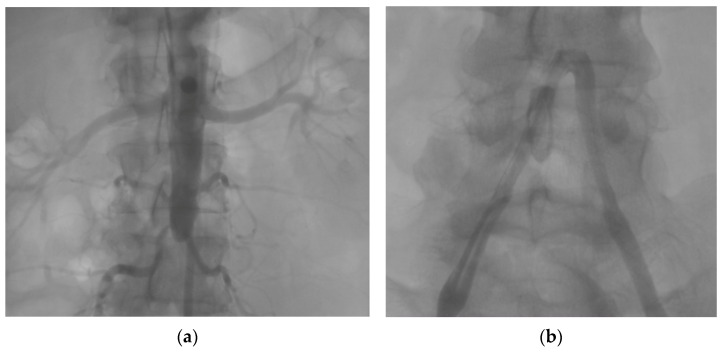
(**a**) Right radial access via pigtail catheter showing complete chronic infrarenal aortic occlusion; renal arteries without lesions—normal opacification; spinal branches; and one uterine branch with normal opacification (collateral circulation). (**b**) Right femoral access via multipurpose catheter—common and external iliac arteries with 40–50% atheromatous plaques diffuse infiltration (or underperfused); no opacification of the aorta prior to the aortic bifurcation suggests complete distal aortic occlusion. Arteriography images of the distal abdominal aorta of a 42-year-old female presenting to the Internal Medicine Clinic with bilateral lower limb pain, which proved to have typical intermittent claudication features, are discussed herein. The onset of symptoms was approximately five years prior to admission. At that time, the patient was investigated with electromyography, which had a normal result, and with venous Doppler ultrasound that indicated chronic venous insufficiency. No improvement of the symptoms was registered after the initiation of diosmin and hesperidin treatment. During the last two years, paresthesia of the lower extremities added to the initial symptoms. A lumbar spine MRI showed only moderate osteoarthritis changes, with no spinal stenosis and no disco-radicular conflicts. The patient—a smoker of 15 pack-years—was known to have mixed dyslipidemia, and her body mass index fell within class I obesity values. At the age of 19, she had a pregnancy complicated by pre-eclampsia, but with a full-term birth, and was consequently diagnosed with essential arterial hypertension after the exclusion of secondary high blood pressure causes. Her blood pressure has been kept under control ever since with angiotensin-II-receptor antagonist therapy (candesartan). The medical history of the patient also included one spontaneous abortion in the ninth week of pregnancy and one fetal death in the twelfth week of pregnancy. The absence of bilateral peripheral pulses (of the dorsalis pedis artery, posterior tibial artery, popliteal artery, and femoral artery) was noted during the physical examination. The performed ankle-brachial index had an equal bilateral value of 0.54, suggestive of severe peripheral arterial disease. The patient had a complete blood count within normal range (8200 white blood cells/mm^3^, 3390 lymphocytes/mm^3^, 269,000 platelets/mm^3^, and 14.2 g/dL hemoglobin), normal levels of inflammatory markers (erythrocyte sedimentation rate 18 mm/h and C reactive protein 3 mg/L—normal level < 5 mg/L), with dyslipidemia being the only alteration of blood tests (total cholesterol 205 mg/dL and triglycerides 168 mg/dL). Considering the obstetric history of the patient, an immunological work-up was carried out. The results indicated negative antinuclear antibodies, negative anti-double-stranded DNA antibodies, negative anti-beta-2-glycoprotein antibodies, negative anti-La antibodies, negative anti-Ro antibodies, negative anti-Sm antibodies, negative lupus anticoagulant factor, negative anti-cardiolipin IgG antibodies, weakly positive anti-cardiolipin IgM antibodies (24 MPL/mL, normal values < 20 MPL/mL), and negative antineutrophil cytoplasmic antibodies (c-ANCA, p-ANCA). An arterial Doppler exam of lower limb vessels showed diffuse changes in arterial waveform and a lack of Doppler signal in the anterior tibial artery, bilaterally. No atheroma, stenoses or thromboses were found. The patient was referred to the cardiology clinic for further investigation. She underwent arteriography, and the result emphasized the presence of a five-centimeter complete infrarenal aortic occlusion that did not involve the aortic bifurcation (this figure). The presence of collateral arteries between the lesser pelvis and the internal and external iliac arteries was noticed. The angiography could not demonstrate an atherosclerotic or thrombotic etiology of the complete distal aortic occlusion.

**Figure 2 diagnostics-13-01346-f002:**
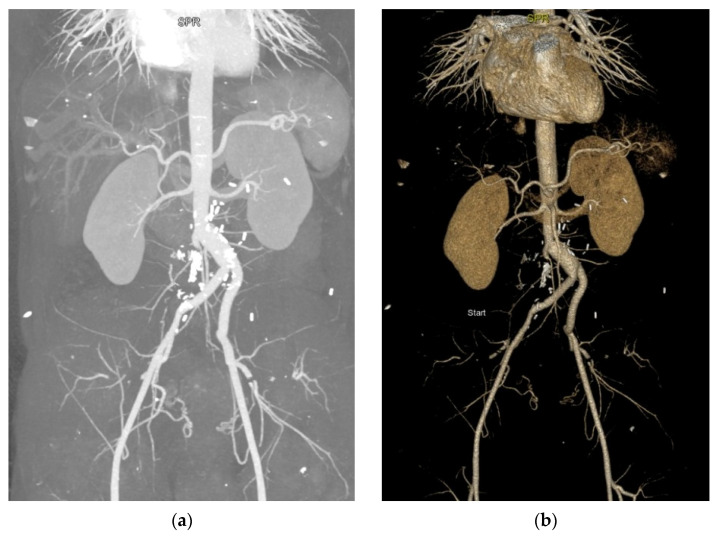
(**a**) Post-aortoiliac bypass arteriographic image; diffuse arterial calcifications; and (**b**) 3D reconstruction of arteriographic image of aortoiliac bypass. The cardiologist recommended thrombophilia re-evaluation. The screening for acquired thrombophilia—including anti-cardiolipin IgM and IgG antibodies and lupus anticoagulant factor—was negative. Protein C, protein S, homocysteine and antithrombin levels were tested, with no significant changes detected. Factor V Leiden and prothrombin gene mutation were absent. Thus, the inherited thrombophilia diagnosis could not be sustained. The patient was redirected to the vascular surgery department for revascularization. At discharge, antiplatelet therapy (acetylsalicylic acid, 75 mg daily), antihypertensive therapy (candesartan, 16 mg daily), proton pump inhibitor therapy (pantoprazole, 20 mg daily), statin therapy (atorvastatin, 10 mg daily), and cilostazol (100 mg daily) were prescribed. The patient underwent aorto-iliac Dacron graft bypass surgery (this figure). The postoperative evolution was favorable. The patient experienced digestive discomfort under antiplatelet therapy with 75 mg acetylsalicylic acid tablets. In consequence, augmentation to a new dosage form of 100 mg tablets was decided. This pharmacological product ensured the same antiaggregant effect, but with better gastrointestinal tolerability, due to the different pharmaceutical excipients found in the composition. Statin therapy was increased to 80 mg atorvastatin daily for two months, then 40 mg daily for an indefinite period. Antihypertensive therapy with 16 mg candesartan daily was maintained.

**Figure 3 diagnostics-13-01346-f003:**
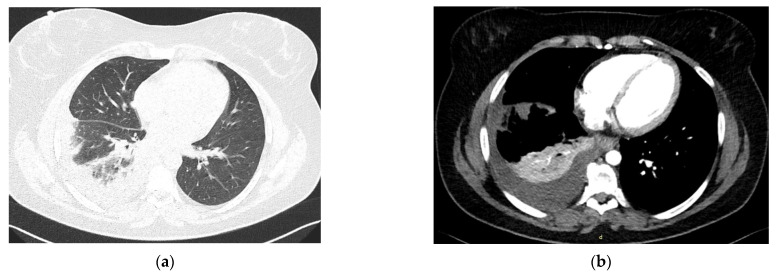
Thoracic CT scan images showing (**a**) pulmonary embolism, pulmonary infarction, and (**b**) secondary pleurisy. Approximately fifteen days after surgery, the patient developed right thoracic stabbing pain, fever, and chills. She was admitted to the cardiology clinic for investigations. The patient was hemodynamically stable. Pulses were present in the lower limb arteries, and the Doppler ultrasound exam found no signs of venous thrombosis. Blood tests showed leukocytosis, elevated levels of inflammatory markers, and a very high D-dimer level. Right heart cavities systolic dysfunction was excluded via cardiac ultrasonography. Chest radiography revealed discrete opacities located in the right lung’s base and opacity of the costo-diaphragmatic right recess. A thoracic contrast-enhanced CT was performed, and the result indicated a pulmonary embolism located in the right lower lobe, pulmonary infarction, and secondary right pleurisy (this figure). Despite the initiation of anticoagulant therapy with unfractionated heparin—therapeutic doses—and antibiotic therapy with ceftriaxone, the clinical state of the patient deteriorated, and she developed sepsis. A new computed tomography showed progression of the thrombus and of the pulmonary parenchymal lesions. Difficulties were encountered in monitoring the unfractionated heparin anticoagulant treatment using activated partial thromboplastin time (aPTT). A fixed dose alternative anticoagulant therapy was researched. In addition, blood tests indicated leukocytosis, very high levels of inflammatory markers, an elevated procalcitonin level and hepatic dysfunction. In consequence, anticoagulant therapy was changed to fondaparinux—therapeutic doses (7.5 mg daily, the dose corresponding to a 70 kg weight)—that ensured lower hepatic toxicity compared to low molecular weight heparins. Acetylsalicylic acid, meropenem and vancomycin were associated in the treatment, with significant clinical and paraclinical improvement. The patient was discharged with an acenocoumarol anticoagulant therapy recommendation. At a subsequent assessment, screening for thrombophilia was performed once more. Lupus anticoagulant factor was positive and remained positive after more than twelve weeks, apart from the first result. Taking into consideration the two positive determinations of lupus anticoagulant factor (more than twelve weeks apart), the transient weakly positive anti-cardiolipin IgM antibodies, the obstetric history of the patient, and the occurrence of pulmonary embolism, the cardiologist established an antiphospholipid syndrome (APS) diagnosis. Oral anticoagulant therapy with acenocoumarol was prescribed indefinitely. No other arterial or venous thrombotic events in any vascular territory were registered under anticoagulant therapy after five years of follow-up. APS belongs to the group of acquired thrombophilias [1]. This systemic autoimmune disorder is mainly characterized by vascular thrombosis in the venous, arterial, or small vessel territory, or by the existence of pregnancy-related morbidity, in association with the persisting presence of lupus anticoagulant factor, anticardiolipin antibodies, or anti-beta-2-glycoprotein antibodies [2]. The presented case brings emphasis to a rare pathology—complete distal aortic occlusion. The most frequent etiology of aortoiliac occlusive disease seems to be atherosclerosis. The emerging site of the lesions is most often represented by the distal aorta and the origin of common iliac arteries. In such cases, the development of collateral circulation to supply the lower limb arterial rete is not unusual [3]. Although our patient had some notable risk factors for atherosclerosis—dyslipidemia, high blood pressure, smoking, and obesity [4]—her young age was probably the most important misleading feature, given the fact that the risk increases after age 45 in men and only after age 55 in women [4]. The gradual development of symptoms over nearly five years and the presence of diffuse arterial calcifications—without causing significant stenoses—on the arteriography images (Figure 2) also suggested a possible atherosclerotic etiology of the occlusion. Cases of complete aortic occlusion due to vasculitis have been described in the literature [5,6]. However, our patient had no clinical characteristics of Takayasu arteritis or giant cell arteritis. When large vessel vasculitis is suspected in a young female patient, Takayasu arteritis is the most common differential diagnosis that should be investigated. Despite this, no vascular bruits were detected via auscultation in any of the large arteries in the case of our patient. No pulse reduction in the arteries of upper limbs or carotid arteries was found during physical examination. No significant difference in systolic blood pressure between arms was noted. Laboratory investigations showed none of the characteristic alterations for large vessel vasculitis [7]—no inflammation, and no significant changes in the complete blood count. Moreover, except for the aortic occlusion, no other vessel wall anomalies were observed in any of the evaluated arteries on the arteriography, which is considered a gold standard investigation in the diagnosis of Takayasu arteritis [8]. Thus, the vasculitis diagnosis could not be sustained. It is well known that thrombophilias are a major risk factor for the development of both arterial and venous thromboses [9]. In particular, APS has been associated with acute aortic occlusion [10]. Although the vast majority of sources report acute thrombotic events in relation to thrombophilias, cases of chronic arterial—and even aortic—thromboses in APS patients have been reported [11,12]. Thrombotic etiology cannot, therefore, be excluded in the case of our patient. Although the etiology could not be certainly established, considering the available data, the complete aortic occlusion of our patient had, most probably, a mixed mechanism of occurrence, implying atherosclerosis and chronic thrombosis of the vessel. The atypical age of onset of the atherosclerotic pathology for a female patient caused a delay in the diagnostic process. Data regarding the association of pulmonary embolism and abdominal aortic occlusion in thrombophilias are scarce. Although postoperative state and hospitalization are known as risk factors [13], the involvement of thrombophilia in the occurrence of the pulmonary embolism cannot be excluded. Thus, the thrombotic event could possibly represent a supplementary argument for the APS diagnosis, in addition to the paraclinical criteria and the obstetric history of the patient. The presented case brings attention to a rare pathology. The scarcity of data regarding thrombophilia and atherosclerosis association that can be found in the literature—consisting mainly of case reports—only serves to confirm the uncommonness of this etiology of aortic occlusion. The obtained results are limited to a single case, and they cannot be extrapolated to other patients with similar symptomatology. However, the presented results may increase awareness of the rare causes of arterial occlusion. Even though in young patients, who present with intermittent claudication symptoms, vasculitis represents the most common etiology, it is always recommended to consider less frequent possible causes—such as atherosclerosis or thrombophilias—in the differential diagnosis.

## Data Availability

Not applicable.

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
