# Peer review of "Chronic Complete Distal Aortic Occlusion and Pulmonary Embolism—Atypical Antiphospholipid Syndrome?"

_diagnostics, 2023, doi:10.3390/diagnostics13071346_

Round 1

Reviewer 1 Report

Journal

Diagnostics

Title

 Chronic Complete Distal Aortic Occlusion and Pulmonary Embolism – Atypical Antiphospholipid Syndrome?

Avery interesting case and elegant presentation

18.. 20..  Further research regarding the involvement of 18 thrombophilias in chronic arterial thromboses and the interaction between thrombotic pathology 19 and other coexisting morbidities – such as atherosclerosis – is required. Moreover, supplementary 20 data on rare etiologies of aortic occlusion are needed….it will be better t o add a conclusion rather than a recommendation at the end of the abstract

More details are required about other laboratory routine investigations like CBC …etc.

The paragraphs are too long and need to be divided into smaller ones.

How was Takayasu arteritis excluded?

Give strength, limitation and recommendation regarding the manuscript.

Author Response

Point 1: 18.. 20..  Further research regarding the involvement of 18 thrombophilias in chronic arterial thromboses and the interaction between thrombotic pathology 19 and other coexisting morbidities – such as atherosclerosis – is required. Moreover, supplementary 20 data on rare etiologies of aortic occlusion are needed….it will be better t o add a conclusion rather than a recommendation at the end of the abstract

Response 1: We thank the Reviewer for this feedback. The highlighted fragment has now been rewritten as suggested (lines 21-24).

Point 2: More details are required about other laboratory routine investigations like CBC …etc.

Response 2: Values for the most significant laboratory parameters at the admission have now been provided in the manuscript (lines 54-58).

Point 3: The paragraphs are too long and need to be divided into smaller ones

Response 3: The structure of the text has now been revised, and the length of the paragraphs has been reduced.

Point 4: How was Takayasu arteritis excluded?

Response 4: Although our patient is a female, and she was 37 years old at the onset of symptoms, no clinical chracteristics relevant to Takayasu arteritis could be found. No vascular bruits were detected by auscultation in any of the large arteries. No pulse reduction in the arteries of upper limbs or carotid arteries was found in physical examination. No significant difference in systolic blood pressure between arms was noted. Levels of the inflammatory markers maintained within normal range. Arteriography is considered a gold standard investigation in the diagnosis of Takayasu arteritis. Both right radial access and right femoral access were used in the case of our patient, due to the presence of the complete distal aortic occlusion. No suggestive angiographic changes were observed in any of the evaluated arteries on the path of the catheter – right radial artery, right brachial artery, right axillary artery, right subclavian artery, brachiocephalic artery, aortic arch, and thoracic aorta. Moreover, Takayasu arteritis rarely affects the distal aorta in the infrarenal segment. Thus, the location of the lesion makes the large vessel vasculitis diagnosis even more improbable in the case of our patient.

Clarifications regarding Takayasu arteritis differential diagnosis have been added in the manuscript (lines 160-170).

Point 5: Give strength, limitation and recommendation regarding the manuscript.

Response 5: Our manuscript attempts to bring attention to a rare pathology. The scarcity of data regarding thrombophilia and atherosclerosis association that can be found in the literature – consisting mainly of case reports – only comes to confirm the uncommonness of this etiology of aortic occlusion. Our results are limited to a single case, and they cannot be extrapolated to other patients with similar symptomatology. However, we hope that the obtained results will increase awareness of the rare causes of arterial occlusion. Even though in young patients, who present with intermittent claudication symptoms, vasculitis represents the most common etiology, it is always recommended to consider less frequent possible causes – like atherosclerosis or thrombophilias – in the differential diagnosis.

Strengths, limitations and recommendations have now been added in the manuscript, as suggested (lines 186-193).

Reviewer 2 Report

Dear Authors

It is possible to provide ABI value

Why antiplatelet therapy dose was upgraded ? and statin only for 2 months ? 

Which heparin was used? LMWH or UFH ? curative or propylactic dosage ?

Why change for fondaparinux ? Which dosage ? What was patient wheight ?

Authors must mentionned that hospitalization is a major risk factor of PE

Line 128 : correct etiOlogy

Author Response

Point 1: It is possible to provide ABI value

Response 1: We thank the Reviewer for this feedback. The right and left ankle-brachial index of our patient had an equal value of 0,54. The exact value of the index has been added in the manuscript (line 53).

Point 2: Why antiplatelet therapy dose was upgraded ? and statin only for 2 months ? 

Response 2: The patient had difficulties in digestively tolerating the 75 mg pharmaceutical product. In consequence, the vascular surgeon decided to upgrade the antiplatelet therapy to the usage of a new dosage form of 100 mg acetylsalicylic acid tablets – one tablet daily. This product ensured the same antiaggregant effect, but proved to have a better gastrointestinal tolerability when compared to the initial prescription, due to the different pharmaceutical excipients found in the composition.

Regarding statin therapy, the patient received 80 mg atorvastatin daily for two months’ time. Subsequently, the dosage was reduced to 40 mg daily, with the recommendation of maintaining it indefinitely.

The paragraph addressing therapy at discharge, after the bypass surgery, has been revised, in order to clarify the highlighted issues (lines 101-106).

Point 3: Which heparin was used? LMWH or UFH ? curative or propylactic dosage ?? 

Response 3: Unfractionated heparin in therapeutic dosage was used at the initiation of anticoagulant therapy.

Heparin type and dosage have been clarified in the manuscript (line 122).

Point 4: Why change for fondaparinux ? Which dosage ? What was patient wheight ? 

Response 4: Difficulties were encountered in monitoring the unfractionated heparin anticoagulant treatment by using activated partial thromboplastin time (aPTT). Despite reducing the unfractionated heparin doses, the aPTT values maintained elevated, hardening the assessment of anticoagulation intensity. Thus, a fixed dose alternative anticoagulant therapy, with a more predictable pharmacokinetic profile was searched. In addition, the clinical state of the patient deteriorated, and she developed sepsis. Blood tests indicated leukocytosis, very high levels of inflammatory markers, elevated procalcitonin level and hepatic dysfunction. Under these circumstances, anticoagulant therapy was changed to fondaparinux, that seems to have lower hepatic toxicity when compared to low molecular weight heparins. Therapeutic doses were used. The weight of the patient was 70 kilograms, thus the required therapeutic dose was 7,5 mg daily.

The reasons for choosing fondaparinux and the used dose has been clarified in the manuscript (line 125-133).

Point 5: Authors must mentionned that hospitalization is a major risk factor of PE

Response 5: The paragraph regarding possible etiologies of the pulmonary embolism has been revised and the importance of hospitalization and postoperative state as risk factors in the pathology of pulmonary embolism has been mentioned (lines 181-185).

Point 6: Line 128 : correct etiOlogy

Response 6: The spelling and grammar of the manuscript have now been revised. The suggested correction has been performed.

Round 2

Reviewer 1 Report

The manuscript improved to a great extent